# PeerJ

# *Bacillus anthracis* pXO1 plasmid encodes a putative membrane-bound bacteriocin

Agata Perlińska[1] and Marcin Grynberg[2]

[1] Centre of New Technologies, University of Warsaw, Banacha, Warsaw, Poland
[2] Institute of Biochemistry and Biophysics PAS, Pawińskiego, Warsaw, Poland

## ABSTRACT

Evolutionary advantages over cousin cells in bacterial pathogens may decide about the success of a specific cell in its environment. Bacteria use a plethora of methods to defend against other cells and many devices to attack their opponents when competing for resources. Bacteriocins are antibacterial proteins that are used to eliminate competition. We report the discovery of a putative membrane-bound bacteriocin encoded by the *Bacillus anthracis* pathogenic pXO1 plasmid. We analyze the genomic structure of the bacteriocin operon. The proposed mechanisms of action predestine this operon as a potent competitive advantage over cohabitants of the same niche.

## INTRODUCTION

Antimicrobial peptides have been found in most living organisms. They play important roles in the innate immunity to protect their hosts from invading pathogens. Such peptides are widely produced by bacteria. Individual bacterial cells compete with both close and distant cells, with other strains and species, to survive under conditions of food shortage. Bacteria produce two types of antimicrobial peptides: ribosomally synthesized peptides (bacteriocins) which inhibit mainly closely related bacteria, and nonribosomally synthesized peptides which inhibit more distant species (*James, Lazdunski & Pattus, 2011*). Bacteriocins are grouped into four classes (I–IV) based on their biochemical and genetic properties. Class I and II bacteriocins are small membrane peptides that kill other bacteria either by producing cell membrane channels or by inhibiting peptidoglycan biosynthesis (*Brötz & Sahl, 2000*; *Fimland et al., 2005*). Class III bacteriocins are high molecular mass proteins ($>30$ kDa) that either degrade cell walls or disrupt the membrane potential. Class IV is usually considered to include all remaining bacteriocins, i.e., complex, modified peptides with diverse targets, although some researchers define class IV as encompassing only cyclic bacteriocins (*Riley & Chavan, 2006*).

*Bacillus anthracis* is an infective agent of cattle and humans. Due to its highly resistant spores, it has been used as a biological weapon, which makes studies of this species especially important. Bacteriocins, key players in intraspecies warfare, deserve special interest. The non-redundant (nr) database at NCBI lists 6 families of bacteriocins identified in *Bacillus anthracis*. However, of these, only one family of thiazole-containing

Corresponding author
Marcin Grynberg,
greenb@ibb.waw.pl

heterocyclic bacteriocins, called heterocycloanthracins (*Riley & Chavan, 2006*), has been characterized. This *B. anthracis* bacteriocin is encoded by a chromosomal gene.

In this work we present the results of our analysis of the previously uncharacterized BXA0138-BXA0140 operon from the pXO1 plasmid. The pathogenicity-related plasmids pXO1 and pXO2 carry most of the armor of *B. anthracis*. However, the functions of many proteins encoded by these plasmids are still unknown. Considering their significance for human health, it is important to identify the functions of all pXO1- and pXO2-encoded genes. What we find here is that the analyzed operon encodes a new and atypical cyclic-like bacteriocin located close to the oedema factor (CyaA). It is as short as all cyclic bacteriocins and is encoded on an operon together with two genes that code for putative attachment and adherence proteins.

## MATERIALS AND METHODS

To identify sequences similar to BXA0138, BXA0139 and BXA0140, we made queries at the National Center for Bioinformatics Information (NCBI) using PSI-BLAST (*Altschul et al., 1997*) on the non-redundant database (nr). The nr database was used as a benchmark to select a threshold value for further searches with Jackhmmer from the HMMER3 package (http://hmmer.janelia.org/) (Table S1). Twilight-zone hits were analyzed with HHpred, Phyre and FFAS (*Jaroszewski et al., 2011*; *Kelley & Sternberg, 2009*; *Söding, Biegert & Lupas, 2005*). As a result, we applied an E-value of 0.01 and performed iterative hmmer (Jackhmmer) searches. The search converged in 4 iterations. All multiple sequence alignments were done using Clustal Omega from the EMBL-EBI Tools (http://www.ebi.ac.uk/Tools/msa/clustalo/).

The three-dimensional model of BXA0140 was automatically built with the Phyre program using the structure of a *Clostridium difficile* bacteriocin, thuricin CD (PDB code 2LA0), as template (*Sit et al., 2011*). Strong sequence similarity between the template and the analyzed protein is present only in the C-terminal part (see Fig. 1). For this reason, we applied molecular dynamics to properly fold the protein. We used NAMD (*Phillips et al., 2005*) to obtain a stable model of the protein using NPT ensemble (constant pressure and temperature). The simulation was performed for 25 ns in 310 K (minimization and molecular dynamics done alternately) until the stable structure, based on C-alpha carbon RMSD, was reached. We used CHARMM36 force field and set the time step to 1 fs. The protein was placed in the center of a water box with 10 Å layer from the structure and with periodic conditions. In order to stabilize the system NaCl ions were added. Correctness of the model was validated using PROSESS (*Berjanskii et al., 2010*). The structure was visualized with PyMOL (http://www.pymol.org/).

The Xre binding site was found using the DBTBS database searching tool (http://dbtbs.hgc.jp/) (*Sierro et al., 2008*). When searching a sequence with a PSSM, the score of each subsequence is calculated by adding the contribution of each base in the subsequence according to its position and to the PSSM. To determine if the scored subsequence could be a TFBS (transcription factor binding site), a *p*-value threshold is used. The *p*-value here is the probability for a random sequence to have an equal or higher score. When selecting

A.

```
BXA0138          LPKGVTFVGEPSKLNKEKLKLQYYVKGYVRNAGEAYVKIIVDLRKDSNVGR---I-
Adhesin repeat1  LNDGLNFVGDKGQVIQKKLNETLAIKGNLDAAaVVTDK---NLRVDNDKDKNGELI
Adhesin repeat2  LNDGLNFVGDKGQVIQKKLNETLAIKGNLdANATVTDK---NLRVDNDKDQNGELI
Adhesin repeat3  LNDGLNFAGNQGDTIVKKLNETLTVKGALANTADASSe---NLRVDSQDG---ALV
                 * .*:.*.*: ..   :**:   :** :   . . :   :** *.:      :
```

B.

```
BXA0139 N-terminus  --------MIFTQDGNSLRVESKNNGKSVSNKRHAWSVRVDGKEIYSFHESTKVEQVVDAFNKLPKVKKQGKQFEVVQKSHEIH
BXA0139 C-terminus  RLGSKDGQLTFFKVGSVFSMYAQKPDKSISTKDKNWAVLVDGVIKHTFSKK-TVRDVKASIDKL---KLRGQRFELQL------
B. cereus HlyII     KLNKGKGKLSLSMNGNQLKATSSNAGYGISYEDKNWGIFVNGEKVYTFNEKSTVGNISNDINKL---NIKGPYIEIKQI----
B. antracis HlyII   QLNKGKGKLSFSMNGNQLKATSSNAGYGISYEDKN*GIFVNGEKVYTFNEKTTVGNISNDINKL---NIKGPYIEIKKI----
                    : :  *.:  :.:   .:* : : .: *:*   ::* :. .* ::   ::** : :* :*:
```

C.

```
BXA0140                                   -----------------MKKKHIVTTTALSFGLIA--LGGVGTTFAAADILRNLA-------Q---------
Thuricin CD C-terminus (2LA0)             --------------------------------LA--SGGVGTEFAAA-----------------------
Carnocyclin A (2KJF)                      ---------LVAYGIAQGTAEKVVSLINAGLTVGSIISILGGVTVGLSGVFTAVKAAIAKQGIKKA-----IQL
Bacteriocin AS-48 (1E68)                  MAKEFGIPAAVAGTVLNVVEAGGWVTTIVSILTAVG--SGGLSLLAAAGRESIKAYLK-KEIKKKGKRAVIAW
B. subtilis Subtilosin A leader sequence  -----------------MKKAVIVE---------------------------------------------
```

**Figure 1** **Multiple sequence alignments of (A) BXA0138, (B) BXA0139 and (C) BXA0140 with the identified similar sequences (for details see "Results" section).** (A) Adhesin repeats belong to the *Actinobacillus pleuropneumoniae* autotransporter adhesin [GI: 501454838]. (B) *Bacillus cereus* and *Bacillus anthracis* hemolysin II domains derive from GI: 446632026 and GI: 673614593, respectively. The star shows the nonsense mutation (TGG to TGA), instead of tryptophan 372 in *B. cereus*. The remaining part of the protein is reproduced using BLASTX analysis of the C-terminal part of the *B. anthracis* DNA sequence. (C) Comparison of BXA0140 to thuricin CD, bacteriocin AS-48 and carnocyclin A. PDB IDs are given in parentheses. Multiple sequence alignment also shows similarity of the leader sequence to the leader of *Bacillus subtilis* subtilosin A. Coloring of alignments is a default of the Clustal Omega program.

a threshold of 1%, all the subsequences for which this probability is below 0.01 will be returned possible TFBS.

## RESULTS

Following the procedures described in the method section, we identified distant similarities of the three proteins encoded on the BXA0138–BXA0140 operon to known protein families (see alignments in Fig. 1).

The C-terminal part of BXA0140 is similar to a bacteriocin called thuricin CD (Probab = 94.86, E-value = 0.014 in HHpred search), but the N-terminus and the middle of the sequence lack cysteines or methionines which are required to build sulfur bridges (Figs. 1 and 2) (*Sit et al., 2011*). BXA0140 shows also similarity to 2LA0 (PDB code); the similarity reaches 77% identity and encompasses almost half of the protein, with a confidence score of 76.1. Similarity to other bacteriocins in the alignment is mostly based on the hydrophobic pattern where hydrophobic residues are divided by other amino acids, mainly glycines, serines and threonines. BXA0140 is highly hydrophobic, therefore it may be a transmembrane protein.

BXA0140 is a 37 amino acid protein. The first 7 amino acids highly resemble a signal sequence from *Bacillus subtilis* subtilosin A—of these seven residues, five are identical (Fig. 1C). However, when the initial sequence was compared to leader peptides from other bacteriocins, a comparable level of similarity could not be observed. Therefore, the current data does not allow us to tell whether BXA0140 has a signal peptide or not; leaderless bacteriocins have already been reported previously, e.g., enterocin Q or L50 (*Cintas et al., 2000*). In our simulation, the protein was assumed to be leaderless, and all 37 amino acids were modeled.

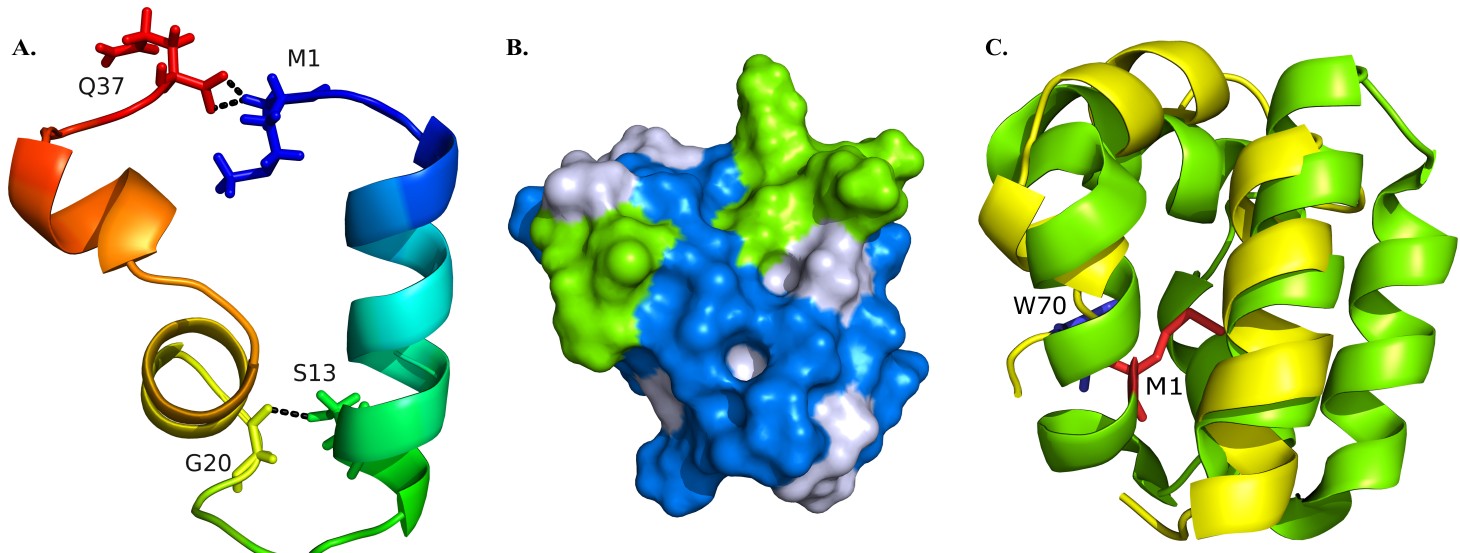

**Figure 2 Structural model of BXA0140.** (A) Representation of the protein colored from N-terminus (blue) to C-terminus (red) using a rainbow color gradient; hydrogen bonds are represented as black dashed lines. (B) Van der Waals surface of the protein with hydrophobic residues colored with blue and charged residues colored with green. (C) Superposition of BXA0140 (yellow) and bacteriocin AS-48 (green; PDB ID: 1E68). N- and C-terminal ends of AS-48 are represented as sticks and colored red and blue, respectively.

The resulting BXA0140 model contains three alpha helices, two of which form a small helix-turn-helix motif. Due to its hydrophobic character the protein tends to form a ring with the side chains of hydrophobic residues directed towards the center and with the N- and C-terminal ends in close proximity to each other (Fig. 2A). The protein ring is stabilized by hydrogen bonds between the two ends. Thuricin CD, which was used as a template for the modeling of the structure of BXA0140, includes thioether bridges in its structure—a common feature among bacteriocins. Although the BXA0140 structure does not contain thioether bridges, we nevertheless think it probably represents a bacteriocin: not only because of its similarity to thuricin CD but also because it contains so many hydrophobic (nearly 63%) and charged (over 16%) residues (Fig. 2B). The location of the charged residues, which are concentrated near the N-terminus of the largest helix (at pH 7 the overall charge of this region is 3.1), may result in the protein being attracted to the negatively charged bacterial membranes (*Montalban-Lopez et al., 2012*). The hydrophobic character of the protein could then allow it to dock into the membrane (Fig. 2B).

The BXA0140 sequence and model structure show also similarity to other bacteriocins, such as AS-48 or carnocyclin A (Figs. 1C and 2C). Bacteriocin AS-48 is a 70-amino acid long circular protein with its C- and N-termini linked by a peptide bond. The "head-to-tail" linkage in AS-48 was the first example of this kind of post-translational modification (*Samyn et al., 1994*). Despite the difference in length, BXA0140 shows both sequence similarity (30% identical and 30% similar residues; Fig. 1C) and structure similarity (shared helix-turn-helix motif; Fig. 2C) with AS-48 (PDB ID: 1E68). Superposition of these proteins shows that the C- and N-terminal ends of BXA0140 are positioned in the helix-turn-helix motif in a similar manner as the "head-to-tail" linkage of AS-48.

Another of the analyzed proteins, BXA0138, is similar to BXA0149 (pXO1-117), a protein encoded just 10 genes away on the same pXO1 plasmid. A Jackhmmer search also reveals similarity of BXA0138 to the autotransporter adhesin from *Actinobacillus pleuropneumoniae* serovar *13* str. N273, but this similarity is below the assumed threshold. The central part of the *Bacillus anthracis* protein is similar to each of the 3 duplicated parts of the adhesin (Jackhmmer scores from 3.4 to 9.5, e-values from 0.056 to 0.00071). These fragments of the *A. pleuropneumoniae* adhesin are in turn similar to the head domains of the known adhesin from *Bartonella henselae* that binds the extracellular matrix, BadA (HHpred: Probab = 99.89, E-value = 5e−23) (*Szczesny et al., 2008*).

The third analyzed protein, BXA0139, displays homology to the C-terminal end of hemolysin II, a $\beta$-barrel pore forming toxin ($\beta$-PFT) from *Bacillus cereus* (*Miles, Bayley & Cheley, 2002*). This homology has already been described by *Miles, Bayley & Cheley (2002)*, but only as encompassing a 46-amino acid segment of BXA0139. What we find here, however, is that BXA0139 contains a duplication of this fragment, and both the N- and C-terminal parts of BXA0139 are similar to the C-terminus of hemolysin II (Fig. 1). The significance of the C-terminus of hemolysin II in *B. cereus* is unknown; functional studies suggest it has no influence on the hemolytic activity of the enzyme (*Baida et al., 1999*; *Miles, Bayley & Cheley, 2002*). Hemolysins typically form heptameric rings (*Gouaux, Hobaugh & Song, 1997*; *Song et al., 1996*), in which the C-terminal domains reside in the outer parts of the monomers (*Miles, Bayley & Cheley, 2002*). *Kaplan et al. (2013)* hypothesize "that the C-terminal domain, henceforth denoted as HlyIIC, facilitates the attachment of the toxin to cell membranes".

Additionally, we also searched the nucleotide sequence of the investigated operon for potential regulatory motives, and we identified a putative Xre transcription repressor site about 100 bp upstream of the BXA0140 open reading frame. The Xre repressor blocks the transcription of a phage-like bacteriocin, PBSX (phibacin damaged-prophage) (*McDonnell & McConnell, 1994*; *Wood, Devine & McConnell, 1990*). The predicted Xre-binding site sequence is GATACAAAGAGTAAA—very similar to the consensus GATACATTTTGTATC (with a DBTBS score 7.30139).

## DISCUSSION

In this brief note we demonstrate the striking similarity of a pXO1 plasmid operon to known elements of bacterial toxins and of attachment/adherence systems (Fig. 3). We propose here a novel view on the BXA0138-140 operon, suggesting that it forms part of *Bacillus anthracis* warfare.

In the BXA0140 sequence we found a strong thuricin CD-like motif. Taken together with the appropriate length and hydrophobicity of the protein, this leads us to the conclusion that BXA0140 most likely represents a bacteriocin. Its circular structure can only be inferred from sequence similarity to other circular bacteriocins, from the arrangement of hydrophobic and charged residues and from the identification of possible hydrogen bonds that could stabilize the closed structure. In the simulation the protein tends to maintain its circular structure which suggests that it may either represent a form

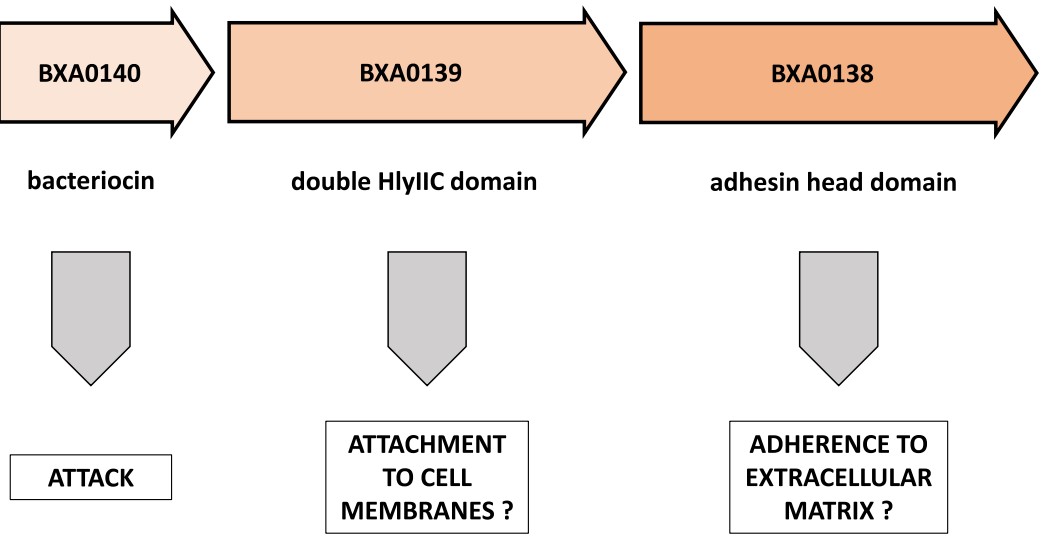

**Figure 3 Putative functions of proteins encoded by the BXA0138-BXA0140 operon.** This model predicts a combination of functions that are related to competition between bacterial strains.

switching between linear and circular bacteriocins or that the closed structure forms later, in post-translational modification. BXA0140 also contains a helix-turn-helix motif similar to that of the AS-48 bacteriocin, which suggests a similar function and similar mechanism of binding to bacterial membrane. However, it is also possible that BXA0140 has a 7-amino acid leader peptide, basing on the similarity of the initial sequence to the signal sequence from *Bacillus subtilis* subtilosin A. This notion is supported by the fact that both *B. anthracis* and *B. subtilis* belong to the same *Bacillus cereus* group. If this is the case, then the mature form of BXA0140 would lack N-terminal fragments necessary for formation of the circular structure.

Analysis of the BXA0139 sequence—which was found to show similarities to *B. anthracis* BXA0149, *A. pleuropneumoniae* adhesin and BadA adhesin—suggests that it is a putative extracellular matrix binding protein. In this operon it would give an advantage to its owner in stabilization and survival of the strain.

The last analyzed protein, BXA0138, was shown to have homology to parts of *B. cereus* hemolysin II. The interpretation of this result requires further research, because two contradictory hypotheses can be proposed. First, BXA0138 could be a helper protein for hemolysin II in host attachment. Second, it could be a tittering protein that impairs the functioning of hemolysins from neighboring bacteria. Also, it is possible that the main monomer domain performs some auxiliary function, maybe as a regulatory domain.

We are aware that our results yield only suggestions for further analysis and that other hypotheses on the structure and function of the investigated proteins cannot be excluded. However, there are many examples of borderline homology detections that were later proven correct, e.g., the prediction showing that DUF185 belongs to the S-adenosyl-L-methionine-dependent methyltransferase fold (SAM-Mtase) (*Sadreyev et al., 2007*), confirmed later by crystallography (PDB ID 1zkd).

We note the absence of immunity proteins encoded in this operon. This is clearly contradictory to the traditional view about the location of toxin and immunity genes in immediate vicinity. However, according to Zhang and colleagues, the *Bacillus cereus* group possesses a large excess of immunity genes over toxin genes (*Zhang et al., 2012*), which clearly implicates that the pairing concept can not hold in all cases. Therefore the lack of immunity proteins in the BXA0140-0138 cluster does not necessarily indicate a different, not toxicity-related function of the cluster, nor does it mean that there exist no immunity genes directed against this particular toxin, especially directed against non-kin strains.

Our results yield a number of novel hypotheses that deserve experimental investigation. This can be extremely challenging since the function(s) of the BXA0138-0140 operon may be subtle and deletions will not necessarily produce clear phenotypical differences. However, the identification of BXA0140 as a putative novel bacteriocin deserves particular attention, since every new antimicrobial agent represents a hope for a new group of antibiotics.

## ACKNOWLEDGEMENTS

We thank Drs. Anna Muszewska, Marta Hoffman-Sommer, Krzysztof Pawłowski and Anthony Pugsley for reading and reviewing our manuscript.

### Funding
The authors declare there was no funding for this work.

### Competing Interests
The authors declare there are no competing interests.

### Author Contributions
- Agata Perlińska performed the experiments, analyzed the data, wrote the paper, prepared figures and/or tables.
- Marcin Grynberg conceived and designed the experiments, performed the experiments, analyzed the data, wrote the paper, prepared figures and/or tables, reviewed drafts of the paper.

### Supplemental Information
Supplemental information for this article can be found online at http://dx.doi.org/10.7717/peerj.679#supplemental-information.

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
