# Peer review of "Bacillus anthracis pXO1 plasmid encodes a putative membrane-bound bacteriocin"

_PeerJ, doi:10.7717/peerj.679_

## Round 0.1 · original submission · Major Revisions

I see the importance of this MS concerning a putative membrane-bound bacteriocin produced by Bacillus anthracis pXO1 plasmid. Please improve this MS as suggested by the reviewers.

·

Basic reporting

The article is generally well-written and clear.

Minor comments:
1. The English should be checkd, with attention to punctuation (some missing comas), usage of has vs have, and usage of plural vs singular.
2. Citing Wikipedia should be avoided in favour of scientific literature.
3. The discussion of "absence of the immune proteins" is unclear to the reviewer.

Experimental design

The work has been properly conducted.

Minor comments:
1. The NAMD method name should be explained, a nd authors should explain what they minimised.
2. When citing BXA0140 similarity to c2la0A, authors should explain what algorithm was used.

Validity of the findings

The arguments presented are, in general convincing. However, in Discussion, the authors should admit that similarities they detect are sometimes of borderline statistical significance. Perhaps they could cite examples of studies where such "borderline" homology prediction has been proven right.

Additional comments

This is an interesting piece of work, and after addressing the minor comments above does deserve publication.

·

Basic reporting

The manuscript is clearly written

Experimental design

No Comments

Validity of the findings

In line 66, 82 the PDB Id should be 2LA0 (Thuricin CD) What is given is not correct.
In line 81 ,82 the details regarding the alignment of the BXA0140 sequence with the Thuricin CD sequence used for modelling are not correct. The alignment shown in Fig 1 has only 17-32 of BXA010 with 14-29 of Thuricin CD and shows only 5 identical residues. This is not 77% identical nor does it cover half the sequence. This has to be corrected.
The model building and dynamics is questionable owing to the lack of alignment and the possible membrane bound nature of the molecule. However, this does not invalidate the other results of the manuscript. Either the model could be left out or there could be a statement mentioning that this model is only representative.

Reviewer 3 ·

Basic reporting

In the manuscript ‘Bacillus anthracis pXO1 plasmid encodes a putative membrane-bound bacteriocin’ the authors claim regarding the discovery of a putative membrane-bound bacteriocin encoded by the Bacillus anthracis pathogenic pXO1 plasmid. This claim is mainly based on bioinformatics tools, including sequence alignments and homology modeling. The work is of interest; nevertheless, the presentation is poor in the present form and it needs sizeable improvement before this manuscript should be considered for final publication in PeerJ. The language is poor at several places and it will be good for the authors to take some help from a language expert to correct the grammatical errors and further improve the paper.

Experimental design

The authors need to mention whether the present manuscript is a full paper or a short communication. This reviewer understands the limitation of the present work, neverthless, due to lack of necessary expertimental evidence, this manuscript demands sufficient explanations as to why the claim regarding the discovery of a putative membrane-bound bacteriocin is valid (which at present seems a bit superficial and is mainly based on the analysis of the genomic structure of the bacteriocin operon). There seems to be some gaps in Materials and Methods, Result and Discussion sections, and as presented, they fail to provide complete and concise information, wherever they are due.

Validity of the findings

In Figure 1, the amino-acids are written with small letters (not clearly visible). It will be good to increase the size of the one-letter code in Figure1. Further, the sequence alignment is based on only single bacteriocin sequence (for BXA0140 with thuricin CD). This raises concerns about the comparision with bacteriocin being superficial than specific. The authors should also provide a comparative alignment with at least few more known bacteriocins (both cyclic and linear ones).

Further, the figure legends lacks clarity and should provide more information.

·

Basic reporting

Authors have used suite of bioinformatic approaches to ascribe biological function to operons BXA0138 - BXA0140. Authors propose that the cyclic like putative bacteriocin biosynthesis operon on the plasmid gives competitive advantage to Bacillus anthracis in a gut ecosystem.

Here are the specific areas that need attention:
1) Line 47: details missing on what features make this bacteriocin unusual.

2) Line 26 - 29: reference missing

3) Line 32: wikipedia page has been cited; better reference required.

Experimental design

1) Top jackhummer hits of the operon should be summarized in a table.

Validity of the findings

1) Bacteriocin biosynthesis is accompanied with signal peptides/ pepitades. (Reference: Manuel Montalbán-López et al; J. Biol. Chem. 2012, 287:27007-27013 Discovering the Bacterial Circular Proteins: Bacteriocins, Cyanobactins, and Pilins).

Presence or absence of such elements in or around operon BXA0138 - BXA0140 has not been highlighted.

2) Authors highlight that the presence of biosynthesis operon BXA0138 - BXA0140, gives a competitive advantage to the Bacillus strain.

Are there any evidence of horizontal gene transfer or plasmid stability indicating that the operon has been acquired in recent past to serve competitive advantage?

---

## Round 0.2 · accepted · Accept

Thank you for your submission to PeerJ. This manuscript is accepted for publication, after reviewers were agreed. Congratulations!!!

·

Basic reporting

A tiny correction:
on page 10, line 158:
is "BadA adhesion", should probably be "BadA adhesin"

Experimental design

No comments

Validity of the findings

No comments

Additional comments

No comments

·

Basic reporting

Corrections mentioned in the earlier review have been carrid out.

Experimental design

Corrections indicated earlier carried out

Validity of the findings

No comments

Additional comments

No comments

·

Basic reporting

No comments

Experimental design

No comments

Validity of the findings

No comments

Additional comments

The revised manuscript is considerably improved and addresses the concerns raised by this reviewer.